# Spatiotemporal Variations in the Water Quality of Qionghai Lake, Yunnan–Guizhou Plateau, China

**Jiao Ran [1], Rong Xiang [1,2,*], Jie Li [1], Keyan Xiao [3] and Binghui Zheng [1,*]**

1    State Key Laboratory of Environmental Criteria and Risk Assessment, Chinese Research Academy of Environmental Sciences, Beijing 100012, China
2    School of Environment, Tsinghua University, Beijing 100084, China
3    Xichang Environmental Monitoring Station, Xichang 615000, China
*    Correspondence: xsxiangrong@163.com (R.X.); zhengbinghui@craes.org.cn (B.Z.)

**Abstract:** Although Qionghai Lake is one of the 11 large and medium-sized lakes (lake area > 25 km$^2$) in the Yunnan–Guizhou Plateau (YGP), there has been little research on its water quality, especially over the long term. Herein, meteorological, hydrologic, trophic, and biochemical indices were investigated over the 2011–2020 period to explore the spatiotemporal variations in water quality in Qionghai Lake. The results showed that the *CCME-WQI* value for Qionghai Lake ranked between marginal and fair during 2011–2020, that the water quality of Qionghai Lake before 2017 was worse than after 2017, and that the water quality of the western part of Qionghai Lake was worse than that of the eastern part. Total nitrogen and total phosphorus were 0.39–0.51 and 0.019–0.027 mg/L during 2011–2020, respectively, and were the main pollution factors in Qionghai Lake. In addition, Qionghai Lake was at the mesotrophic level, but the chlorophyll and trophic state levels (*TLI*) increased year by year, and the levels in the western area were higher than in the eastern area. Increased anthropogenic activities (industrialization, urbanization, agricultural intensification, etc.) were the main reasons for the poor water quality of Qionghai Lake before 2017, while, after 2017, effective government environmental restoration and management measures improved the water quality. Moreover, the difference in land-use types within the watershed was the main reason for the spatial heterogeneity of water quality in Qionghai Lake. Potassium permanganate index (COD$_{Mn}$) and ammonia nitrogen content index (NH$_3$-N) were not very high, but both showed seasonal variations. Water transparency (SD) in Qionghai Lake was reduced by sediment input and increased algal biomass, while dissolved oxygen (DO) decreased due to thermal stratification. This study is expected to provide a theoretical reference for understanding changes in the water quality and water environmental protection of Qionghai Lake and the YGP.

**Keywords:** water quality; eutrophication; spatiotemporal variation; Yunnan–Guizhou Plateau; Qionghai Lake

## 1. Introduction

Lakes are some of the most important freshwater ecosystems in the world and are suffering from various environmental pressures [1,2]. Since the 20th century, with the unprecedented acceleration of industrialization, urbanization, and agricultural intensification, the influx of nutrient and pollution loads from excessive human activities have exacerbated the pollution and degradation of lake environments [3–5]. Messina et al. [6] found that watershed development stimulated water quality decline in Lake Auburn, ME, USA. Meanwhile, studies have shown that the water temperature (WT) of most lakes worldwide has increased rapidly in the past 30 years at a rate of about 0.3 °C per decade [7,8]. Furthermore, the annual average temperature increase in inland China was 1.38 °C from 1951 to 2009, and it is expected to increase by 2.3–4.6 °C by the end of this century [9]. Wang et al. [10]

pointed out that rising WT exacerbated severer cyanobacterial blooms in Dianchi Lake during 1990–2015. Therefore, anthropogenic pressures and climate change have made water quality deterioration a significant problem in lake water environments worldwide [11,12].

There are 2693 lakes in China, covering an area of 81,414.6 km$^2$, and about one-third of them are freshwater lakes [11]. However, studies have shown that lake water quality has declined significantly in recent decades [2,12]. Severe eutrophication and algal blooms have occurred frequently in the plain lakes in the middle and lower reaches of the Yangtze River [13–16], which pose a serious threat to water quality and limit the development and health of society [13]. The Yunnan–Guizhou Plateau (YGP) lake area is one of the five lake regions in China and contains many rift lakes with small watershed areas and fragile aquatic ecosystems [17]. Studies have shown that climate warming and human activities have altered water quality in some YGP lakes [18,19].

Huang et al. [4] found that human activities (gross domestic product, population, land-use cover, and urban wastewater discharge) aggravated the water eutrophication of Dianchi Lake from 1977 to 2009 in the YGP. In addition, rising temperatures promoted the release of nitrogen and phosphorus from sediments in this lake [20], while a large amount of non-point source load from farmland was the main reason for the frequent occurrence of cyanobacterial blooms in Erhai Lake [21,22]. Zhou et al. [23] pointed out that the apparent decrease in water transparency of Fuxian Lake over the years is related to the increase in potassium permanganate index and air temperature. However, research on the water quality of the YGP lakes has mainly focused on these three lakes, while studies of the other lakes remain rare.

Qionghai Lake is one of the 11 large and medium-sized lakes (lake area > 25 km$^2$) in the YGP and the second largest lake in Sichuan Province. As an important water conservation area in the upper reaches of the Yangtze River, a series of environmental problems are a cause for concern. Qionghai Lake was in a mesotrophic state during 2002–2011, and total phosphorus (TP) and total nitrogen (TN) were the main pollution factors [24]. Zhang [25] found that the TN, TP, and organic pollutant concentrations in the Guanba river estuary of Qionghai Lake showed an increasing trend over time. Additionally, the Qionghai Basin is dominated by easily weathered and eroded sandstone and is located in an earthquake zone prone to mountain torrents that lead to serious soil erosion [26]. Soil erosion has caused sediment to be deposited into the lake at a rate of 17.09 mm/a, which threatens to significantly shorten the life of Qionghai Lake [27]. However, few studies, especially based on long time series, have been conducted on variations in the water quality of Qionghai Lake and on the relevant driving factors.

In the present study, based on work carried out over 10 consecutive years, meteorological conditions, hydrology, water quality indicators, nutrient indices, and land-use patterns were measured from 2011–2020. The specific objectives of this study were to (1) illustrate the temporal and spatial variation in the water quality parameters and determine the key pollution factors in Qionghai Lake, (2) evaluate the trophic level and its spatiotemporal variation in Qionghai Lake, and (3) investigate variation in human activities, meteorology, and hydrology and analyze their impact on the water quality of Qionghai Lake. This study can provide a reference for understanding the water quality variations and formulating scientific water environment management measures for Qionghai Lake and the YGP.

## 2. Materials and Methods

### 2.1. Study Area

Qionghai Lake (102°07′~102°23′ E, 27°47′~28°01′ N) is located in Xichang City, Sichuan Province, China. It is a suburban, semi-enclosed lake situated in the upper reaches of the Yalong River, a tributary of the Jinsha River, and belongs to the Yangtze River Basin. The geography of the Qionghai Basin consists of mountains, hills, and lake basins, with large vertical differences. The Qionghai Lake has a predominantly subtropical plateau monsoon climate, and the monsoon usually runs from June to September [26]. Some general information about Qinghai Lake is summarized in Table 1.

**Table 1.** General information on Qinghai Lake.

| Natural Parameters | Value | Units | Sources of Data |
|---|---|---|---|
| Catchment area | 307.67 | km$^2$ | |
| Lake area | 27.41 | km$^2$ | |
| Eastern/western lake area | 25.12/5.29 | km$^2$ | |
| Lake storage capacity | $2.93 \times 10^8$ | m$^3$ | |
| Eastern/western lake storage capacity | 2.58/0.35 | m$^3$ | Xichang |
| Mean lake depth | 10.95 | m | Environmental |
| Eastern/western lake depth | 12.69/2.94 | m | Monitoring Station |
| Maximum lake depth | 18.32 | m | |
| Mean annual precipitation | 1030 | mm | |
| Average temperature | 17 | °C | |
| Mean humidity | 61.4 | % | |

### 2.2. Data Sources and Processing

All surface water samples were collected monthly in 500 mL plastic bottles with a water depth of 0.5 m and taken to the lab to measure water quality and eutrophication indicators during 2011–2020. The 11 sampling sites for Qionghai Lake were investigated in this study, including QH1, QH2, QH3, QH4, QH5, QH6, QH7, QH8, QH9, QH10, and QH11 (Figure 1). The overall dataset included 288 samples from 4 sampling sites (QH2, QH5, QH8, and QH9) for 2011–2016, and 528 samples from all 11 sampling sites for 2017–2020. Additionally, the monthly sewage treatment data in this study were obtained from the Ecological and Environmental Protection Bureau, the Environmental Monitoring Station, and the Qionghai Administrative Bureau of Xichang City. The datasets for the economy, population, land use, precipitation, and air temperature from 2011 to 2020 in the Qionghai Basin were obtained from the Statistical Yearbook of China Urban Construction, the Statistical Yearbook of Sichuan Province, and the Statistical Bulletin of Xichang National Economic and Social Development.

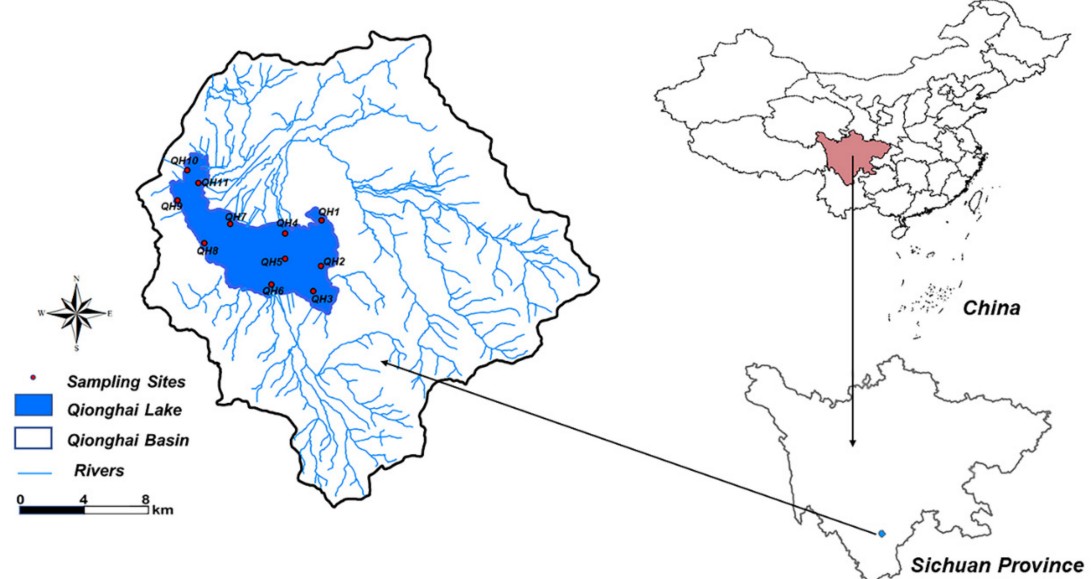

**Figure 1.** Sampling sites for Qionghai Lake.

### 2.3. Analytical Methods

#### 2.3.1. Spatiotemporal Analyses of Water Quality

The water quality parameters and their determination methods are listed in Table 2. Chlorophyll a (Chl-a) was extracted with 90% acetone and measured according to Nusch [28]. Lake water transparency was measured on-site using a Secchi disc (SD) [29]. The comprehensive trophic state level (*TLI)* was calculated using the concentrations of total ni-

trogen (TN), total phosphorus (TP), potassium permanganate index ($COD_{Mn}$), SD, and Chl-a [22,30]. The detailed computational equation for *TLI* was used in accordance with previous studies [31]. *TLI* values varied from 0 to 100 and were divided into five levels: oligotrophic (<30), mesotrophic (30–50), lightly eutrophic (50–60), moderately eutrophic (60–70), and highly eutrophic (>70).

**Table 2.** The water quality indices used and their abbreviations, units, standard values for China's Environmental Quality Standard for Surface Water (GB) Class II, and measurement methods [32].

| Parameter | Abbreviation | Units | GB Class II Standard Value | Method |
|---|---|---|---|---|
| Pondus hydrogenii | pH | | 6–9 | GB6920-86 |
| Dissolved oxygen | DO | mg/L | 6 | GB11913-89 |
| Potassium permanganate index | $COD_{Mn}$ | mg/L | 4 | GB11892-89 |
| Ammonia nitrogen content index | $NH_3$-N | mg/L | 0.5 | GB7479-87 |
| Total nitrogen | TN | mg/L | 0.5 | GB11894-89 |
| Total phosphorus | TP | mg/L | 0.025 | GB11893-89 |
| Water temperature | WT | °C | | GB13195-91 |
| Suspended solids | SS | mg/L | | GB 11901-1989 |

Usually, spatial interpolation analysis with a Geographic Information System (GIS) uses a known value to estimate the value of unknown loci, and it is commonly used in geology, hydrology, and meteorology [33,34]. The inverse distance weighting method (IDW) is a common spatial interpolation analysis method that is used to predict unknown simulated points based on statistical and mathematical methods when a series of sampling points are known [33,35]. Firstly, the IDW assumes that each measurement point can produce a local effect. Secondly, it assumes that measurement points that are closer to each other are more similar than ones that are farther apart [33,36]. Generally, the closer to the known sample point, the greater the weight assigned to the unknown. In this study, ArcMap version 10.8 and IDW were used to estimate the spatial distribution of water quality indices in Qionghai Lake.

In addition, cluster analysis (CA) was used to analyze the spatial distribution of water quality in Qionghai Lake. The sites were classified according to the similarity of the selected water quality indices at each site, and the sites with strong similarities were grouped into a cluster [37,38]. In the present study, 3 physical indicators, including SD, WT, and SS; 6 biochemical indicators, including TP, TN, $COD_{Mn}$, $NH_3$-N, pH, and DO; and 2 nutrient indices, including Chl-a and *TLI*, were used for cluster analysis.

2.3.2. Mann–Kendall (MK) Test

The Mann–Kendall (MK) test is widely used to directly capture hydrological and climate trends by using a dataset series, regardless of whether the series is linear or non-linear [39–41]. During the detection of trend changes, pre-whitening is not required, but the magnitude of the trend and the length of the series should be considered in detail [42]. Since the MK test can adaptively detect natural change processes using a time series, it is usually used to assess fluctuations in precipitation, water level, water temperature, and water quality indices [42]. Therefore, in this study, the MK test was selected to analyze temporal variations in each water quality indicator, using the 'trend' package in R software vR3.6.1(R Core Development Team, 2019, Vienna, Austria).

According to the results of the MK test, Sen's slope parameter (S) was used to ascertain the increasing or decreasing trend of the selected indicators. Usually, S < 0 indicated a downward trend, while S > 0 showed an upward trend. Furthermore, *p*-values (*p*) represented the significance levels of the indicators' variation trends: $p \leq 0.01$ indicated an extremely significant level, $0.01 < p \leq 0.05$ indicated a significant level, $0.05 < p \leq 0.1$ indicated a slightly significant level, and $p > 0.1$ indicated that the indicator showed almost no variation.

### 2.3.3. Canadian Council of Ministers of the Environment—Water Quality Index (*CCME-WQI*)

The *CCME-WQI* has been used for a comprehensive assessment of the water quality of lakes, rivers, reservoirs, and other natural water bodies in different countries [43,44] because it can provide a flexible indicator template that adapts to water quality measurement under different conditions caused by the heterogeneity and particularity of the natural environment by adjustment of the parameters [45–47]. It can assess whether the water quality meets the specified water quality goals easily but synthetically and indicate the water status category. To explore the water quality of Qionghai Lake, *CCME-WQI* and China's Environmental Quality Standard for Surface Water (GB) Class II [32] were both used. The calculation process accorded with a previous study [45]. According to the score values of *CCME-WQI*, the water quality of the lake was divided into five categories: poor (0–44), marginal (45–64), fair (65–79), good (80–94), and excellent (95–100).

## 3. Results and Discussion

### 3.1. Temporal Variations in the Water Quality Indices

Previous studies have shown that the water quality of Qionghai is better than that of Dianchi and Xingyun Lakes and other heavily polluted plateau lakes, though worse than that of Fuxian and Lugu Lakes [45,48]. As a centralized domestic drinking water source, a habitat for aquatic creatures, and a spawning ground for fish and shrimp, the water quality of Qionghai Lake has always been required to meet GB Class II standards [32].

In the present study, the $COD_{Mn}$, $NH_3$-N, DO, and pH values of Qionghai Lake met the GB Class II values [32] between 2011 and 2020 (Figure 2a), with annual average ranges of 2.07–2.65, 0.087–0.136, 6.7–7.91 mg/L, and 8.08–8.41. However, during 2011–2020, the annual average TN and TP were in the range of 0.39–0.51 and 0.019–0.027 mg/L, and the months exceeding Class II [32] were 26 and 37 (Figure 2a), respectively. Therefore, TN and TP were the key pollution indicators for Qionghai Lake. Moreover, TN and TP concentrations in Qionghai Lake showed an increasing trend from 2011 to 2017, and the monthly average values were in the range of 0.01–0.04 and 0.21–1.23 mg/L (Figure 2a). After 2017, both indices' concentrations decreased with time (Figure 2a).

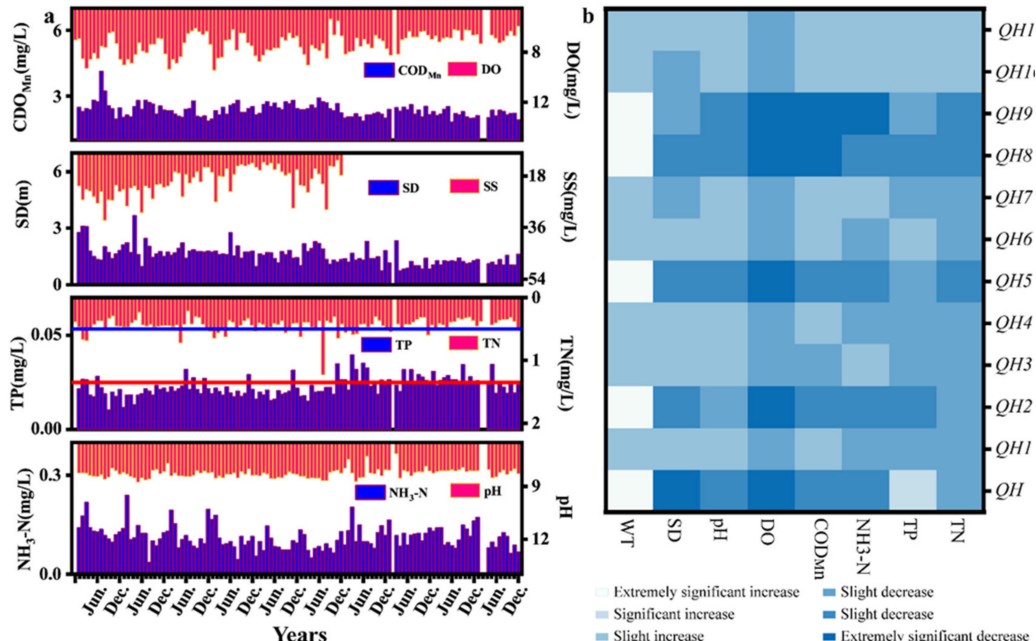

**Figure 2.** (**a**) Variations in water quality parameters for Qionghai Lake from 2011 to 2020. The Class II values for TP and TN were marked as red and blue lines, respectively. (**b**) Mann–Kendall test (MK test) results for water quality parameters of Qionghai Lake.

The elevated TN and TP concentrations between 2011 and 2017 were due to anthropogenic pressures. In the present study, the gross domestic product (GDP), resident population (Pop), and urbanization rate (UR) of the Qionghai Lake Basin increased from 29.68 to RMB 48.14 billion, from 0.734 to 0.778 million, and from 52.68% to 58.92% (Figure 3a), respectively. These three indicators have a well-fitting relationship with the annual average TP and TN values (Figure 3b), indicating that the acceleration of human production activities and social development have contributed to the increase in TN and TP. In addition, the area of cultivated land in the Qionghai Basin was 69.91 km² in 2010, accounting for 21.26% of the total area (Figure 3, Table 3). This result indicated that abundant agricultural activities could generate plenty of nutrient loads and increase TN and TP concentrations in Qionghai Lake [49]. Non-point source pollution promoted TN and TP contents, which were frequently reported in the Yunnan–Guizhou Plateau (YGP) lakes, such as Erhai Lake [50], Xingyun Lake [48], and Yilong Lake [51].

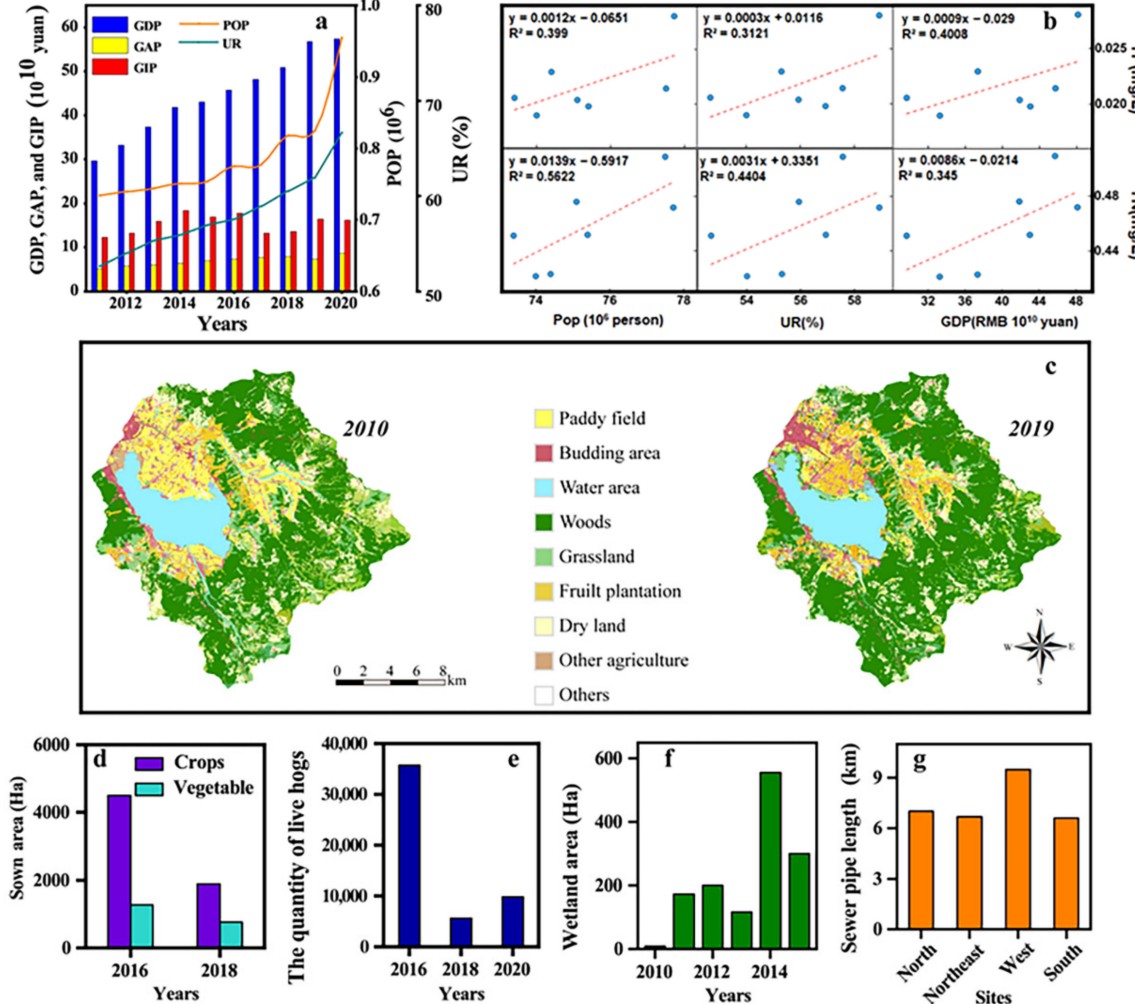

**Figure 3.** (**a**) Gross domestic product (GDP), gross agricultural product (GAP), gross industrial product (GIP), population (Pop), and urbanization rate (UR) of Xichang City from 2011 to 2020. (**b**) The results of linear fitting of TP and TN with GDP, population (Pop), and urbanization rate (UR), respectively. (**c**) Land-use patterns in Qionghai Basin in 2010 and 2019. (**d**) The sown area of crops and vegetables in the Qionghai Basin in 2016 and 2018. (**e**) The year-end quantity of hogs in Qionghai Basin in 2016, 2018, and 2020. (**f**) The new area of lakeside wetland in Qionghai Lake from 2010–2015. (**g**) The sewer pipe length in Qionghai Basin on the northern, northeastern, western, and southern lakeshores.

**Table 3.** Area and areal proportion of various land-use types for Qionghai Basin in 2010 and 2019.

| Years | 2010 | | 2019 | |
|---|---|---|---|---|
| Land-Use Patterns | Areas (km$^2$) | Percentage of Catchment Area | Areas (km$^2$) | Percentage of Catchment Area |
| Cultivated land | 65.91 | 21.26 | 46.88 | 15.12 |
| Fruit land | 12.60 | 4.06 | 21.33 | 6.88 |
| Woods | 149.74 | 48.31 | 175.23 | 56.53 |
| Grassland | 26.50 | 8.55 | 11.02 | 3.55 |
| Building land | 20.19 | 6.51 | 23.61 | 7.62 |
| Water area | 32.01 | 10.33 | 31.34 | 10.11 |
| Others | 3.01 | 0.97 | 0.56 | 0.18 |

However, the annual average of TN and TP showed a downward trend after 2017, which was mainly related to the effective environmental restoration and management of local governments. From 2016 to 2018, the sown area of crops in the Qionghai Basin decreased from 4493.8 to 1888.6 ha, of which the vegetable area decreased from 1271.1 in 2016 to 767.47 ha in 2018 (Figure 3d). Moreover, all large-scale livestock and poultry farms in the Qionghai Basin were closed after 2017, and the year-end quantity of hogs decreased from 35,735 in 2016 to 5635 in 2018 (Figure 3e). The sewage interception project around the lake was completed in 2017 collected domestic and tourist sewage and treated it all in the Qionghai Sewage Treatment Plant (Figure 3g). Up to now, a total of 29.792 km of sewer pipes have been built in the Qionghai Basin (Figure 3g). Furthermore, a total of 1353.25 ha of lakeside wetlands were constructed around Qionghai Lake from 2010 to 2015 (Figure 3f). After stable development, this played a significant role in improving water quality after 2017.

The concentrations of COD$_{Mn}$ and NH$_3$-N in Qionghai Lake were not high from 2011 to 2020, but both showed seasonal variations (Figure 2). From 2011 to 2016, COD$_{Mn}$ was higher in summer and autumn than in winter and spring, which may be related to abundant precipitation and surface runoff carrying a large pollution load into the lake in the monsoon [52]. The study showed that the COD$_{Mn}$ concentration in Qionghai Lake mainly originates from urban domestic emissions, but the defects of the domestic sewer system before 2017 led to sewage overflow during the rainy season [26]. In the Qionghai Basin, the average precipitation in the rainy season is usually 951.3 mm during 2011–2020 (Figure 4a,c). While NH$_3$-N was higher in the spring during 2014–2020 (Figure 2a), this might have been affected by the small inflowing runoff and weak dilution ability of the lake [31]. In addition, low temperatures in spring (Figure 4a,b) reduced microbial activity and slowed down the nitrogen cycling in lake ecosystems, resulting in the accumulation and increase in NH$_3$-N [53].

The MK test results showed that both DO and SD in Qionghai Lake showed extremely significant decreasing trends (S < 0; $p$ < 0.01) from 2011 to 2020 (Figure 2a,b). The SD declined by 0.78 m (Figure 2); the growth and functions of aquatic plants may therefore have been affected due to insufficient light [23,54]. The successive rainfall washed a large amount of sediment into the lake, resulting in the inferior SD of Qionghai Lake in the monsoon period [52]. Furthermore, the resuspension of pollution loads in sediment and increase in algal blooms also contributed to the decrease in SD in Qionghai Lake [55,56].

The DO concentration of Qionghai Lake decreased by 0.96 mg/L from 2011 to 2020 (Figure 2). In 2020, DO was lower than Class II [32] from time to time, and the lowest value was 5.86 mg/L. It is well known that excessive pollutants cannot be rapidly degraded and transformed under lower DO concentrations, which will eventually lead to the degradation of water quality in a lake [57,58]. A decline in DO is usually associated with an increase in water temperature [59]. Generally, thermal stratification and water exchange occur when there is a large temperature difference between the water and air at the surface of a deep lake [60]. Then, a bottom layer of water with hypoxia rises to the surface of the lake, resulting in a decrease in DO concentration in the surface water [58]. Anoxic

lower-layer water rises to the lake surface, leading to a drop in DO in the surface water [60]. During 2011–2020, the water temperature of Qionghai Lake increased significantly by 2.4 °C (S > 0; $p < 0.01$), and the temperature difference between water and air gradually increased (Figure 4e).

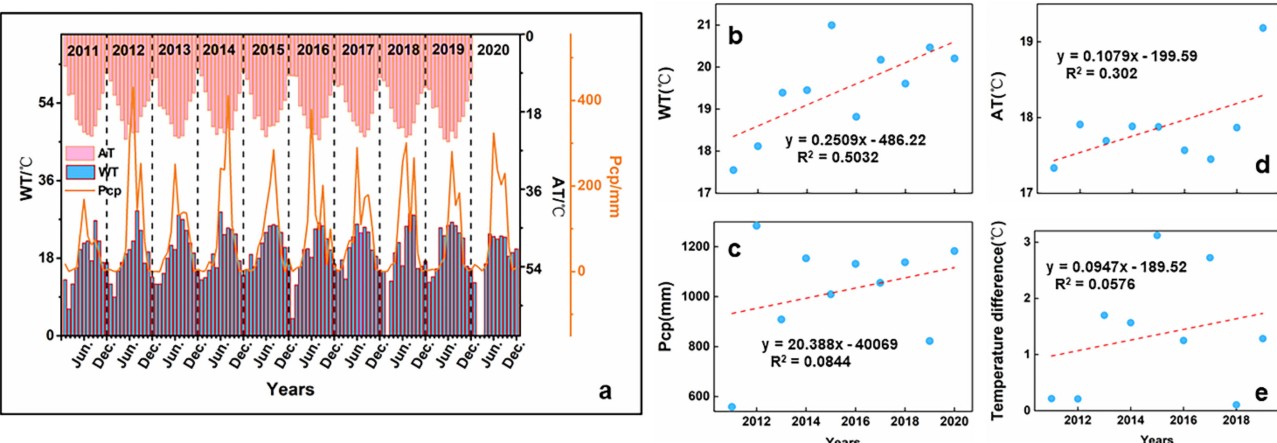

**Figure 4.** (**a**) Monthly average water temperature (WT), air temperature (AT), and precipitation (Pcp) in Qionghai Lake. Annual variation trends for (**b**) WT, (**c**) Pcp, (**d**) AT, and (**e**) the difference between WT and AT.

*3.2. Spatial Variation of Water Quality in Qionghai Lake*

The water quality indices of the 11 sampling sites for Qionghai Lake showed obvious spatial heterogeneity; except for pH, all water quality indicators in the western area were worse than in the eastern area (Figure 5). The mean TP concentrations of QH9, QH10, and QH11 in the western water area were 0.025, 0.047, and 0.065 mg/L (Figure 5h), which values were significantly higher than those of the other sites. The TN concentration of QH11 was 0.57 mg/L, while values for the other 10 sites varied from 0.40 to 0.45 mg/L (Figure 5g). The DO was the highest at QH2 (7.34 mg/L), located in the eastern area, and lowest at QH11 (6.72 mg/L; Figure 5d). The average SD for QH9, QH10, and QH11 in the western water area was less than 1 m (Figure 5c). In addition, spatial variations in the water quality of Qionghai Lake were well illustrated using CA. The water quality of Cluster II (QH10 and QH11) was worse than that of Cluster I (QH1–QH9) (Figure 5i).

In general, the concentration of pollutants in water is jointly affected by human activity intensity and natural environment characteristics [3,61]. Land-use change reflects the impact of human activities on watershed ecosystems [37] and has long been regarded as the main driving force for water quality changes [62]. Wei et al. [63] pointed out that degraded land has had a significant adverse impact on changes in lake/reservoir water quality in China. Chen et al. [37] also found significant relationships between land-use patterns and stream water quality in southern Alberta, Canada.

In Qionghai Lake, different land-use types in the watershed lead to spatial heterogeneity in water quality (Figure 5). The western water area of Qionghai Lake is close to Xichang City, and construction and agricultural land (paddy fields, gardens) were the main land-use types; ecological lands, such as forest and grassland, were less distributed (Figure 3b). The higher proportion of degraded land area, the more the pollution load, while the smaller the ecological land area, the weaker the adsorption and retention capacity of pollutants [63]. Therefore, the western water area of Qionghai Lake suffered more pollution pressure and had worse water quality (Figure 5). Conversely, the eastern water area of this lake had low concentrations of pollutants due to higher vegetation distribution (Figure 5). Similarly, spatial differences in lake water quality caused by land-use change can also be found in other lake basins of the YGP, such as Erhai and Dianchi Lakes [55,64].

Furthermore, the greater the water depth and storage capacity of a lake, the stronger the ability to purify and accommodate pollutants [42]. In the YGP lake area, the mean

depths of Fuxian and Lugu Lakes are 95.2 and 40.3 m; both lakes showed strong self-purification ability and good water quality [62,65]. In Dianchi Lake, Waihai Lake had a larger depth and area than Caohai Lake, and therefore exhibited lower concentrations of pollutants [20]. Regarding the present study, the mean depth of the western part of Qionghai Lake is 2.94 m, and the water area is relatively narrow (Table 1). While the eastern area of the lake is large and deep, the depth of the QH5 can reach 18.32 m (Table 1). Therefore, due to the poor self-purification and anti-interference of the water body, the water quality in the western part of Qionghai Lake was worse than in the eastern part (Figure 5).

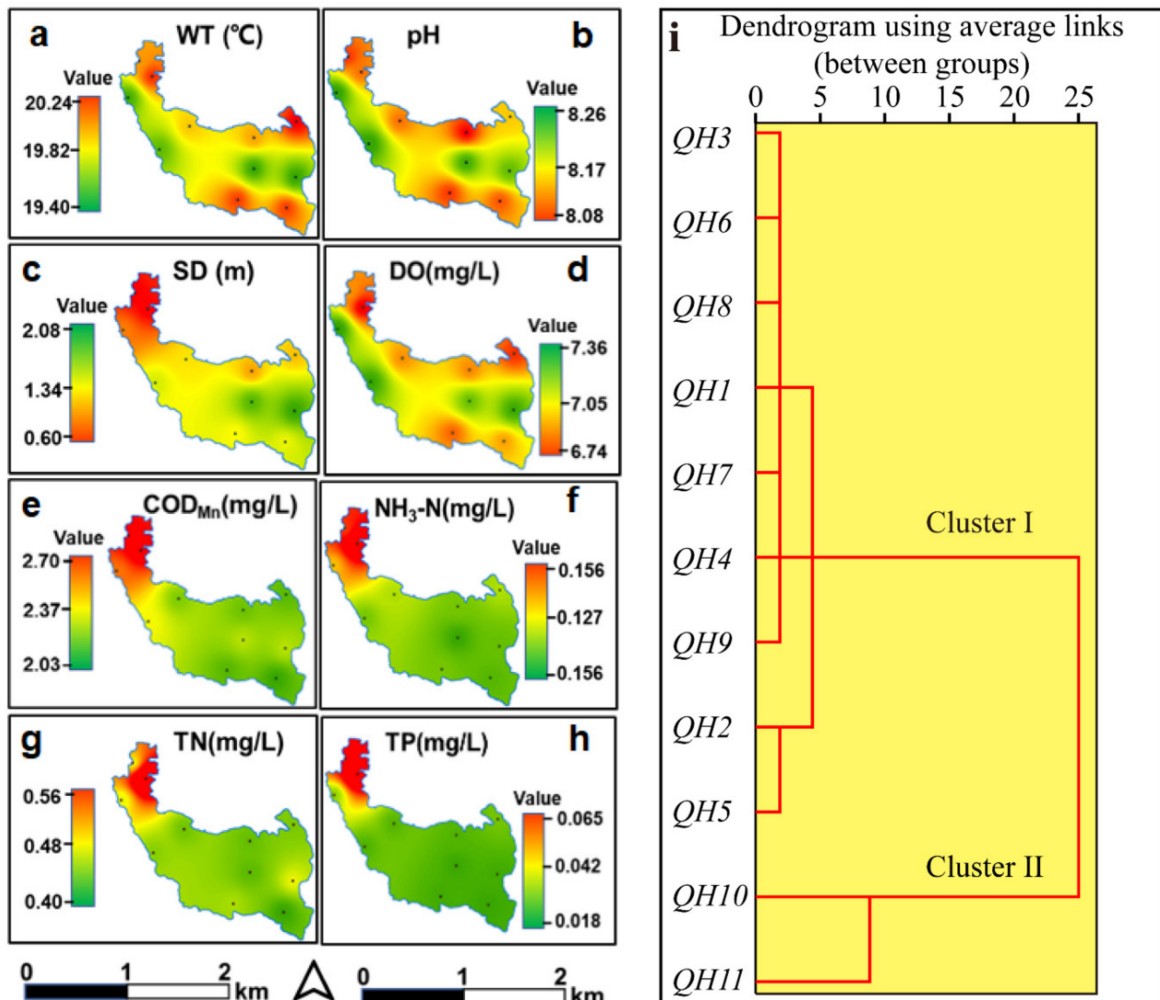

**Figure 5.** Spatial variations in (**a**) WT, (**b**) pH, (**c**) SD, (**d**) DO, (**e**) COD$_{Mn}$, (**f**) NH$_3$-N, (**g**) TN, and (**h**) TP over the 11 sampling sites of Qionghai Lake. (**i**) Dendrogram based on agglomerative hierarchical clustering of the 11 sampling sites for Qionghai Lake.

### 3.3. Eutrophic State and Its Influencing Factors

The Chl-a concentration for Qionghai Lake was lower than the medium nutrient level (≤4 mg/m$^3$) from 2011 to 2015 (Figure 6). However, the Chl-a annual average concentration of Qionghai Lake increased year by year, from 3.59 mg/m$^3$ in 2011 to 8.85 mg/m$^3$ in 2020 (Figure 6a,b). The *TLI* of Qionghai Lake ranged from 30 to 50 and the trophic level of the entire lake was mesotrophic during 2011–2020, and the *TLI* after 2016 was higher than that before 2016 (Figure 6a,c). In terms of spatial distribution, both Chl-a and *TLI* were higher in the western area than that in the eastern Qionghai Lake (Figure 6c). The Chl-a concentration for QH11 was the highest (20.63 mg/m$^3$), followed by QH10 (13.91 mg/m$^3$) (Figure 6c). The *TLI* value for QH11 was the highest in Qionghai Lake, as well (Figure 6c);

the monthly average value varied from 42.82 to 55.46 in 2011–2020 (Figure 6c), reaching a eutrophication state.

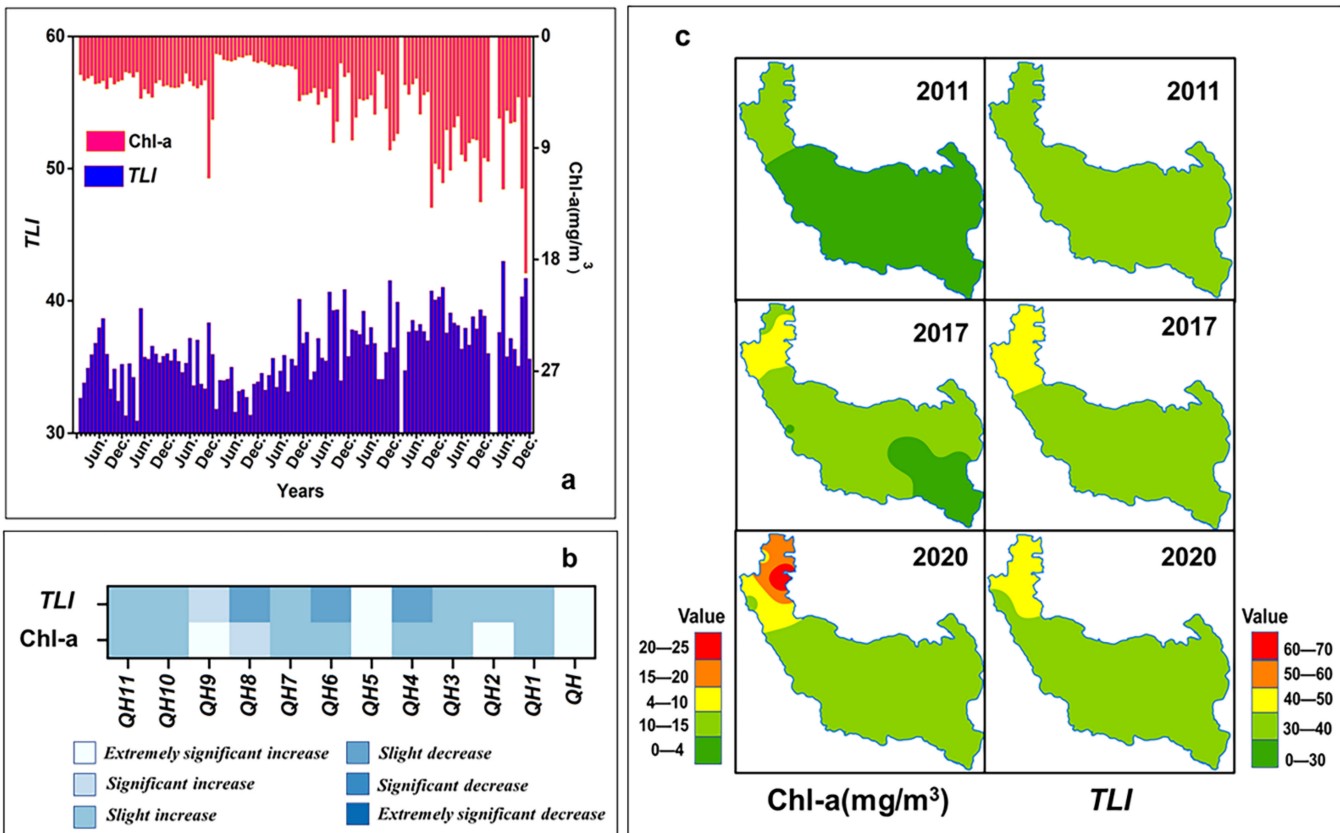

**Figure 6.** (**a**) Temporal variations in *TLI* and Chl-a in Qionghai Lake from 2011 to 2020. (**b**) Mann–Kendall (MK) test results for *TLI* and Chl-a in Qionghai Lake. (**c**) Spatial variations in *TLI* and Chl-a in Qionghai Lake in 2011, 2017, and 2020, respectively.

Generally, excessive N and P concentrations will lead to lake eutrophication [66,67]. Eutrophic water can easily result in the massive reproduction of planktonic algae and the formation of blooms, which ultimately lead to lake ecology degradation and threaten drinking water safety [4,68]. High TN and TP concentrations were one of the reasons for the occurrence of eutrophication in Dianchi and Erhai Lakes [20,69]. In Qionghai Lake, sufficient TN and TP are the material foundations of lake eutrophication (Figure 2). Meanwhile, the TP showed an obvious upward trend ($S > 0$; $0.01 \leq p < 0.05$) during 2011–2020 (Figure 2b), indicating that TP made a great contribution to the increase in the trophic level of Qionghai Lake.

The intensification of human activities and differences in land-use patterns (Figure 3b, Table 3) also contributed to the increase in values of Chl-a and *TLI* for Qionghai Lake. In the western part of Qionghai Lake, a large pollution load enters the water body due to the high intensity of construction and tourism development activities [70,71]. In the eastern area, non-point source pollutants generated by agricultural activities also caused the water body to degenerate from an oligotrophic to a mesotrophic state [22]. The Chl-a concentration for QH11 was the highest and reached the trophic state (Figure 6), showing that this water area was highly polluted and that algal cells proliferated greatly [72].

Additionally, natural environment variations are an important external cause of lake eutrophication and algal blooms [73]. Warming water temperatures (WTs) usually favor phytoplankton reproduction [74], which probably increased the Chl-a concentration and *TLI* levels in Qionghai Lake. SD can affect the growing environment of algae and then influence Chl-a concentration and eutrophic state [23,75]. The SD of Qionghai Lake decreased by

0.78 m extremely significantly (S < 0; *p* < 0.01) during 2011–2020 (Figure 2a,b), indicating that SD was the main physical driving factor behind the rising eutrophication index for Qionghai Lake. Therefore, the trophic level of a mesotrophic lake such as Qionghai Lake may be jointly affected by physical factors, such as SD, and nutrients, such as P [76].

### 3.4. Water Quality Evaluation Based on the CCME-WQI

According to the result of the *CCME-WQI*, the water quality of Qionghai Lake ranked between marginal and fair during 2011–2020, and the annual average value of the *CCME-WQI* was in the range of 64–79 (Figure 7a). The *CCME-WQI* value for the western area was lower than in the eastern area of Qionghai Lake (Figure 7b). QH11 and QH10 showed the lowest *CCME-WQI* values of 45.75 and 55.38, while the *CCME-WQI* mean values of other sampling sites ranged from 67.09 to 80.80 (Figure 7). These results indicated that the water quality at QH11 and QH10 was the worst in Qionghai Lake, which is consistent with the results of the single factor evaluation method (Figure 5). The poorer water quality in the western part of Qionghai Lake was directly exhibited by the higher values of TN, TP, Chl-a, and *TLI* (Figures 2 and 6). These values are possibly related to the dense population, the high intensity of construction and tourism development, and weak water self-purification ability [4,5,77].

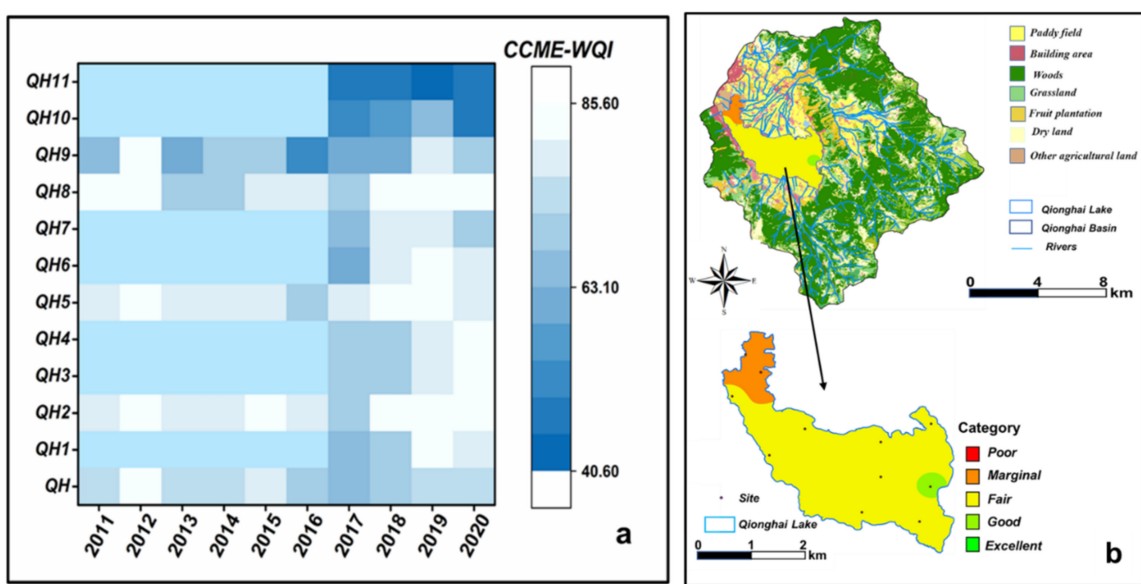

**Figure 7.** (**a**) *CCME-WQI* results with corresponding water quality status for Qionghai Lake during 2011–2020. (**b**) Spatial interpolation analysis based on the *CCME-WQI* values for 11 sampling sites in Qionghai Lake.

Furthermore, in terms of temporal variation, the *CCME-WQI* value for Qionghai Lake decreased from 2011 to 2017 (Figure 7a). This was related to the growth of the economy and the development of urbanization which resulted in a large number of pollution loads entering the lake during this period (Figure 3). Moreover, the increase in water temperature and decrease in SD and DO also exacerbated the degeneration of water environmental quality of Qionghai Lake to some extent (Figure 2). The improved water quality of the lake after 2017 was attributed to government protection and restoration measures (xichang.gov.cn, 9 November 2021).

The single-factor pollution index (SFPI) usually selected the worst water quality index as the comprehensive water quality class [45]. Based on the SFPI method, Gu et al. [24] found that the water quality of Qionghai Lake was classified as Class III during 2002–2011, which was incompatible with the functional requirements of Qionghai Lake. However, the result for *CCME-WQI* obtained in the present study showed that the water quality of Qionghai Lake ranked between marginal and fair (Figure 7a). In general, the *CCME-*

*WQI* assesses general water quality both qualitatively and quantitatively by reference to a group of typical assessment items [47] which can be used to easily determine whether the comprehensive water quality meets the specified water objectives [46]. Therefore, compare to the SFPI method, the assessment results obtained with the *CCME-WQI* were sensible [47] as well as easily understandable and acceptable by the government and public.

## 4. Conclusions

Intensive human activities caused high nutrient loads in Qionghai Lake, TN and TP values varied in the range of 0.39–0.51 and 0.019–0.027 mg/L during 2011–2020, and TN and TP values exceeded Class II for 26 and 37 months. The water temperature increased by 2.4 °C in 2011–2020, leading to an obvious lack of DO in Qionghai Lake. The DO concentration decreased by nearly 1 mg/L and showed an obvious decreasing trend. The decreased SD in Qionghai Lake might be due to exogenous sediment input and endogenous algal proliferation.

The CA and spatial interpolation analysis showed that the water quality in the western area of Qionghai Lake was relatively poor. This might be due to the high intensity of human activities, shallow water depth, narrow water area, lower water storage capacity, and poor water self-purification ability in this region.

The values of Chl-a and *TLI* in Qionghai Lake showed increasing trends from 2011 to 2020, and they were higher in the western area than in the eastern area. Qionghai Lake was at the mesotrophic level, and the increase in *TLI* was mainly caused by changes in SD and TP.

Climate change and human activities caused the spatiotemporal heterogeneity of water quality in Qionghai Lake. The *CCME-WQI* value indicated a marginal to fair ranking during the 2011–2020 period and was lower in the western part of Qionghai Lake. Warming temperatures and increasing urbanization led to the degeneration of the water environment in Qionghai Lake before 2017. After this time, the water quality gradually improved because of effective environmental protection measures.

**Author Contributions:** Conceptualization, J.R., R.X. and B.Z.; methodology, R.X.; software, J.L.; validation, J.R., J.L. and K.X.; formal analysis, J.R.; investigation, K.X.; resources, B.Z.; data curation, K.X.; writing—original draft preparation, J.R.; writing—review and editing, R.X.; visualization, R.X.; supervision, B.Z.; project administration, B.Z.; funding acquisition, B.Z. All authors have read and agreed to the published version of the manuscript.

**Funding:** The research was funded by Joint Research on Ecological Environmental Protection Restoration of the Yangtze River (no. 2019-LHYJ-01-0202).

**Institutional Review Board Statement:** Not applicable.

**Informed Consent Statement:** Not applicable.

**Data Availability Statement:** The datasets used and analyzed during the current study are available from the corresponding authors upon reasonable request.

**Acknowledgments:** The authors acknowledge the support provided by the Ecological and Environmental Protection Bureau, the Environmental Monitoring Station, and the Qionghai Administrative Bureau of Xichang City. We are also grateful for the thoughtful comments provided by three anonymous reviewers.

**Conflicts of Interest:** The authors declare that they have no competing interest.

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
