# Peer review of "Spatiotemporal Variations in the Water Quality of Qionghai Lake, Yunnan–Guizhou Plateau, China"

_water, doi:10.3390/w14152451_

Round 1

Reviewer 1 Report

Reviewing Report

Manuscript Entitled (Spatiotemporal Variations in the Water Quality of Qionghai 2 Lake, Yunnan-Guizhou Plateau, China, from 2011 to 2020) tried to investigate meteorological, hydrologic, trophic, and biochemical indices during 2011-2020 to explore the spatiotemporal variations of water quality in Qionghai Lake, in the Yunnan-Guizhou Plateau (YGP), China. When I review the manuscript, I found that the main topic of the manuscript could be consider within the scope of journal. However, the literature review section needs to be improved greatly and add more modern related works. The language of the whole manuscript need to improve. In addition, there are some point in the whole manuscript need to improve.

Due to these issues, I believe this manuscript is could be consider to for publication in a prestigious journal such as water journal. Below the author/s can find some comments to improve the quality and impact of thier work. I would recommend to rewrite the manuscript addressing the points mentioned.

Abstract

-  The abstract need to rewrite and the authors should focus on the main findings and the main results.

Introduction

-         Page 1- line 31, the authors mentioned (climate change …..) and in the line 32, they mentioned (climate warming…..). The climate change is different from climate warming, therefore, please be careful to use the suitable one.

-         Page 1, line 33, (water quality deterioration has become ……..worldwide. The authors required to explain why this happen.

-         Page 2, line 56, what this symbol (SD) refers to? Please try to use the full name of any term when it is use for first time.

-         Page 2, the paragraph (The annual average temperature in the YGP had increased in the last century [20, 21], …………….. while were rare in other lakes) need to rewrite in order to be more understandable.

-         Page 2, line 64 (As the important ecological protec- 64 tive screen and water conservation area in the upper reaches of the Yangtze River, a series of environmental problems in the Qionghai Lake Basin are cause for concern ) try to mentioned examples of these problem in order to be more clear for the reader.  

-         More previous studies related to manuscript topic in general and specific related to water quality in Qinghai Lake to should be add.

Materials and Methods  

-         It is better to provide the information of Qinghai lake in table.

-         Also, table include the statistical information of the data used should be provided.

-         The methods in table 1 need more explaination.

-         Page 4, first paragraph, it is required to provide more information about the invers distance weighting method and it is better if the information is support by an figure about the example or procedure of this method.

-         Page 4, line 132, the authors mentioned (In In the present study, uses seven biochemical parameters, three hydrological parameters, and two nutrient indexes for cluster analysis), please try to explain what are these parameters? and why only these parameters are used?

-         Why the (CCME-WQI) is used in this study? Why the authors did not used another WQI such National Sanitation Foundation Water Quality Index (NSFWQI), Oregon Water Quality Index (OWQI) and Overall Index of Pollution (OIP)….. etc.

Results and Discussion

-         Quality of the figures must be improved. Some figures can be collected and a new more figures can be added.

-         Page 6, line 91 (These results meant that the 191 human activities in the basin may have greatly contributed to the exceeding TN and TP concentrations). Is this investigated by the current study or from another study because there is reference in the end of sentence? Also, it is required to explain how and why this happened?

-         In general, the results need more explain and discussion where the current discussion dose not enough and the author/s only describe the results without any interpretation.

-         It is recommended to use other Categories of WQI and compare the results with CCME-WQI.

Conclusions

-         The limitation is missing from conclusion section

Reviewer 2 Report

Suggestion for authors is attached

Reviewer 3 Report

General comments

This paper concerns the water quality trend in Qionghai Lake. Many data are shown. Unfortunately, the analysis is not enough. The authors showed many possibilities that affected water quality in the lake. However, there is no direct analysis between water quality change and the possibilities (the authors just pointed out the possibilities and used many "may" and "might."). Moreover, in terms of the research, I could not see any significant research findings which are beneficial for readers. I am afraid to say that this paper is not appropriate for publication as a research paper. Please see the specific comments below.

Specific comments

1.    L. 73–78: The purpose of the study in terms of a new research is not clear. These sentences show the purpose of making a report about Qionghai Lake.

2. L. 101–102: Sampling strategy is not clear. Frequency or date? Depth? Time?  A 500mL bottle for a site, which seems to be a little bit small amount for the analysis.

3. L. 115: Please describe the equation of TLI. 

4. L.121: What is HJ, SL GB/T, and GB?

5. L.132–133: What is 7 biochemical parameters, 3 hydrological parameter, and 2 nutrient indexes?

6. L.163–164: I cannot clearly understand "first decrease and then increase before 2017".

7. L.164: mean is annual average?

8. L.179: Fig.2 is missing Year. 

9. L.192: There is no direct comparison between anthropogenic overexploitation, climate change, population, urbanization, agricultural non-point source, AND water quality change. The author showed only the possibilities. Very weak for the analysis.

10. L.203: Precipitation is a possibility. Please analyze.

11. L.223: Please analyze the DO decrease by temperature increase and other causes.

12. L.247–251: This is a possibility and there is no support analysis.

13. L.255-257: This is a possibility and there is no support analysis.

14. L.322: What is frequent human activities?

Author Response

Please see the attchment

Round 2

Reviewer 1 Report

Please note for the comment related to climate Change / climate warming. I think it is better to use either climate change or global warming. 

Reviewer 2 Report

In the revised form of the manuscript, authors have included majority of suggestions

Reviewer 3 Report

Unfortunately, I could not see any significant progress in research and could not find the meaning of this research. This article can be a report and should not be accepted as a research paper.